# Integrated Machine Learning Approach for the Early Prediction of Pressure Ulcers in Spinal Cord Injury Patients

**DOI:** 10.3390/jcm13040990

**Published:** 2024-02-08

**Authors:** Yuna Kim, Myungeun Lim, Seo Young Kim, Tae Uk Kim, Seong Jae Lee, Soo-Kyung Bok, Soojun Park, Youngwoong Han, Ho-Youl Jung, Jung Keun Hyun

**Affiliations:** 1Department of Rehabilitation Medicine, College of Medicine, Dankook University, Cheonan 31116, Republic of Korea; kimyuna727@dkuh.co.kr (Y.K.); syoungrm@dkuh.co.kr (S.Y.K.); magnarbor@dankook.ac.kr (T.U.K.); rmlee@dankook.ac.kr (S.J.L.); 2Digital Biomedical Research Division, Electronics and Telecommunications Research Institute, Daejeon 34129, Republic of Korea; melim@etri.re.kr (M.L.); psj@etri.re.kr (S.P.); hanhero@etri.re.kr (Y.H.); 3Department of Rehabilitation Medicine, College of Medicine, Chungnam National University, Daejeon 35015, Republic of Korea; skbok@cnuh.co.kr; 4Department of Nanobiomedical Science and BK21 NBM Global Research Center for Regenerative Medicine, Dankook University, Cheonan 31116, Republic of Korea; 5Institute of Tissue Regeneration Engineering, Dankook University, Cheonan 31116, Republic of Korea

**Keywords:** spinal cord injury, pressure ulcer, machine learning, prediction model, laboratory test

## Abstract

(1) Background: Pressure ulcers (PUs) substantially impact the quality of life of spinal cord injury (SCI) patients and require prompt intervention. This study used machine learning (ML) techniques to develop advanced predictive models for the occurrence of PUs in patients with SCI. (2) Methods: By analyzing the medical records of 539 patients with SCI, we observed a 35% incidence of PUs during hospitalization. Our analysis included 139 variables, including baseline characteristics, neurological status (International Standards for Neurological Classification of Spinal Cord Injury [ISNCSCI]), functional ability (Korean version of the Modified Barthel Index [K-MBI] and Functional Independence Measure [FIM]), and laboratory data. We used a variety of ML methods—a graph neural network (GNN), a deep neural network (DNN), a linear support vector machine (SVM_linear), a support vector machine with radial basis function kernel (SVM_RBF), K-nearest neighbors (KNN), a random forest (RF), and logistic regression (LR)—focusing on an integrative analysis of laboratory, neurological, and functional data. (3) Results: The SVM_linear algorithm using these composite data showed superior predictive ability (area under the receiver operating characteristic curve (AUC) = 0.904, accuracy = 0.944), as demonstrated by a 5-fold cross-validation. The critical discriminators of PU development were identified based on limb functional status and laboratory markers of inflammation. External validation highlighted the challenges of model generalization and provided a direction for future research. (4) Conclusions: Our study highlights the importance of a comprehensive, multidimensional data approach for the effective prediction of PUs in patients with SCI, especially in the acute and subacute phases. The proposed ML models show potential for the early detection and prevention of PUs, thus contributing substantially to improving patient care in clinical settings.

## 1. Introduction

Spinal cord injury (SCI) results primarily from traumatic events and can cause considerable sensory and motor impairments and complications [1]. Among the myriad challenges that patients with SCI encounter, pressure ulcers (PUs) are notable; previous studies revealed that over 20% of SCI individuals develop PUs, with significant implications for morbidity, mortality, and quality of life, especially in developing countries [2,3]. Untreated PUs have a significant impact on patient well-being and place a high financial burden on healthcare systems. These ulcers exacerbate physical and emotional distress and reduce patients’ quality of life [4]. Economically, the treatment of PUs is costly, with the U.S. healthcare system spending approximately USD 26.8 billion annually [5].

These PUs typically occur over bony prominences due to prolonged pressure, and key sites for PUs include the sacrum, heels, and ischial tuberosities, with complications ranging from infections to delayed rehabilitation, underscoring the need for early prediction and intervention [6]. Prevention is crucial in the management of pressure ulcers, especially for individuals at higher risk such as those with spinal cord injuries, and regular repositioning, careful skin inspection, and the use of pressure-relieving devices are key strategies [7]. The prediction and early identification of pressure ulcers are vital, as early-stage ulcers can often be managed more easily and heal faster compared to advanced ulcers, thus highlighting the importance of innovative prediction and intervention strategies in healthcare [8]. The severity of SCI varies, with more severe cases, such as complete tetraplegia, showing a higher PU risk due to an extensive loss of sensory and motor functions [9]. This increased risk is attributed to prolonged immobility and areas prone to sores, like the sacrum and heels [10]. The management and prediction of PUs in patients with SCI have advanced through the use of traditional clinical assessments and monitoring tools [11]. While the Braden Scale, Norton Scale, and Spinal Cord Injury Pressure Ulcer Scale (SCIPUS) are commonly used in clinical settings, their predictive accuracy varies considerably among individual patients with SCI, as highlighted in previous studies [12,13]. The Braden Scale evaluates factors like sensory perception and moisture, the Norton Scale focuses on physical condition and activity, and the Spinal Cord Injury Pressure Ulcer Scale is tailored specifically to patients with SCI, considering aspects like spasticity and sweating [12]. In previous studies, the Braden Scale has demonstrated the highest overall accuracy, whereas the Norton Scale has exhibited greater specificity [12,14]. However, another study reported that functional assessments, such as the Functional Independence Measure (FIM), outperformed both the SCIPUS and Braden Scales in terms of accuracy [15]. This variability highlights the need for more individualized and effective assessment tools for pressure ulcer risk assessment in this patient population. Molecular markers, including proinflammatory cytokines like interleukin (IL)-1α, show promise in early pressure ulcer detection, though their clinical application remains exploratory [16,17]. The challenges encompass enhancing predictive accuracy and ensuring that methods are cost-effective, accessible, and universally applicable. Tackling the challenges of pressure ulcer management requires a combination of clinical expertise and cutting-edge technologies, including alternative support surfaces and wireless patient monitoring systems. These technologies are integral for risk identification, effective repositioning, and microclimate control, thereby emphasizing the need for a patient-centric care approach [18].

In recent years, machine learning (ML) has become a significant factor in healthcare, particularly in areas such as diagnosis, prognosis, and personalized treatment [19]. ML uses algorithms, ranging from simple decision trees to sophisticated deep learning models, to uncover complex patterns and correlations, leveraging increased computational power and extensive healthcare datasets [20]. For instance, decision trees use a tree-like structure to represent decisions and their potential outcomes, which makes them highly interpretable and adaptable to different data types [21]. On the other hand, deep learning models employ layered neural networks to analyze data in a complex manner, making them particularly effective at identifying subtle patterns in large datasets [22]. Advanced algorithms are currently being utilized in spinal cord injury (SCI) care to predict neurological and functional recovery through the analysis of medical records and imaging data [22,23]. Machine learning techniques, including these algorithms, are being explored for patients with SCI to identify risk factors for pressure ulcers (PUs) [24]. This addresses the challenge of limited clinical integration due to previously undefined risk factors.

The primary goal of our study is to establish optimal prediction models by comprehensively integrating clinical, physical, and biological parameters, with a focus on improving the accuracy of prognostic predictions for pressure ulcers during the acute and subacute phases of hospitalization in patients with SCI. The secondary goal is to translate these models into practical tools for clinical application to enable the early intervention and effective prevention of pressure ulcers, thereby significantly improving patient outcomes during their hospital stay.

## 2. Subjects and Methods

### 2.1. Ethics and Study Design

This retrospective observational study was approved by the institutional review board (IRB) of Dankook University Hospital (IRB No. 2021-05-021) and was conducted in accordance with the ethical guidelines of the 1975 Declaration of Helsinki. We reviewed the medical records of 1117 patients with SCI from Dankook University Hospital (DKUH) and Chungnam National University Hospital (CNUH) in South Korea. Patients were included if they underwent surgical or conservative treatment for traumatic or nontraumatic SCI with confirmed spinal cord signal changes by spinal magnetic resonance imaging (MRI) as demonstrated in previous studies [25,26] from May 1996 to May 2021. The clinical data during the initial hospitalization period for SCI were collected by three researchers, who were specifically assigned to ensure impartiality and minimize bias. These researchers were not involved in the statistical analysis or development of the ML model due to a separation of roles that was implemented to maintain the objectivity and integrity of both the data collection and analysis phases. The clinical parameters included baseline characteristics, such as sex, age, height, weight, alcohol consumption, smoking status, and medical history; subscale and total score of the International Standards for Neurological Classification of Spinal Cord Injury (ISNCSCI), the Korean version of the modified Barthel Index (K-MBI), and Functional Independence Measure (FIM), which were initially assessed during the initial hospitalization period for SCI. We obtained all the laboratory data from the laboratory medicine department of each hospital. The laboratory parameters included complete blood count (CBC), electrolytes, lipid battery, glucose, albumin, protein, C-reactive protein (CRP), the erythrocyte sedimentation rate (ESR), procalcitonin, blood urea nitrogen (BUN), creatinine, aspartate transaminase (AST), alanine transaminase (ALT), and total bilirubin. The patients with SCI who experienced PUs at least once during the hospitalization period were classified into the PU group, while the patients who never experienced PUs were classified into the non-PU group. For the PU group, we extracted only clinical and laboratory data from 3 days to 60 days before the onset of PU.

### 2.2. Machine Learning Analysis

In this study, the recursive feature elimination (RFE) technique was used for feature selection. In the RFE method, a given machine learning algorithm is trained on the initial set of baseline features, after which the importance of each feature is computed. The least important feature is then iteratively eliminated at each step. This elimination process is repeated until the optimal set of features that contributes significantly to the model remains. The linear support vector machine (SVM_linear) classifier was selected as the training algorithm. The selection process was implemented using the RFE with a cross-validation (RFECV) module provided by the scikit-learn library [27], which ensures robust feature selection by considering the cross-validation performance during the elimination process.

We utilized various machine learning methods, combining advanced deep learning with traditional techniques. We employed the graph neural network–graph convolutional network (GNN-GCN) and deep neural network (DNN), which are complex artificial neural networks. GNN-GCN analyzes data structured in graphs, while DNN processes data through interconnected layers [28]. Traditional methods used for classification include linear support vector machine (SVM_linear) and support vector machine with a radial basis function kernel (SVM_RBF) for decision boundary-based classification, K-nearest neighbors (KNN) for proximity-based classification, random forest (RF) as an ensemble of decision trees to improve prediction accuracy, and logistic regression (LR) for binary outcome probabilities.

The GNN-GCN model was trained using a graph matrix computed by Euclidean distances between input data, while the other models were trained directly on input data. The GNN-GCN and DNN models were implemented using PyTorch 1.10 [29], and the other ML methods were implemented using scikit-learn 0.24.

Fivefold cross-validation was performed to evaluate the model performance. The dataset was randomized and divided into five partitions, one of which was used for testing and the other for training. To ensure balanced case–control ratios in each partition, a stratified K-fold cross-validation method was used. Cross-validation was repeated five times to ensure a robust and reliable evaluation. Model performance was evaluated by accuracy, area under the receiver operating characteristic curve (AUC), and F1-score.

To improve the interpretability of the problem, we performed additional analyses on the decision tree. The decision tree was trained with entropy as the partitioning criterion. The graph shows the use of features for prediction and the corresponding criteria.

### 2.3. Statistics

All the laboratory, neurological, and functional data were compared between the two hospitals using PASW 20.0 (IBM Corp., New York, NY, USA). The Shapiro–Wilk test was used to assess the normality of the distribution of all the numerical data from each group. The chi-squared test was used for categorial parameters, and the independent *t* test was used for continuous parameters to compare the differences between the two groups. *p* < 0.05 indicated statistical significance.

## 3. Results

### 3.1. Flow of the Machine Learning Algorithm

Figure 1 outlines the flow of our machine learning algorithm. Clinical data, sourced from the two participating hospitals, were collected during several preprocessing stages. This involved imputation using the KNN method and subsequent filtering of missing values (NaN). Data from patients with a feature coverage exceeding 80% were subjected to imputation, while those with a feature coverage less than 70% were discarded during the NaN filtering stage. For the PU cohort, data were further curated to capture only the interval, spanning 3 days prior to PU onset, up to 60 days before its incidence. Concurrently, laboratory data pertaining to the PU group were processed through date filtering and imputed using mean values. The processed clinical and laboratory datasets were subsequently combined and applied to feature selection. Using these consolidated data, seven distinct machine learning models were developed. The efficacy of the combinations was determined through a 5-fold cross-validation procedure, with performance metrics presented in terms of accuracy, AUC, and F1-score.

### 3.2. Data Characteristics and Dataset Selection for Each Hospital

Table 1 presents the baseline characteristics of patients from Dankook University Hospital (DKUH) and Chonnam National University Hospital (CNUH), categorized by the presence or absence of pressure ulcers. While DKUH focused on the formulation and refinement of machine learning algorithms, CNUH was utilized exclusively for external validation. Notably, patients with pressure ulcers exhibited longer hospital stays across both institutions. For our machine learning models, we utilized parameters from the ISNCSCI, K-MBI, FIM, and 20 laboratory indicators. A detailed analysis revealed significant differences in certain metrics. Within the ISNCSCI, total motor scores, especially for the right and left lower extremities, showed marked disparities. Sensory scores, both light touch and pinprick, varied notably between groups. According to the K-MBI metrics, distinctions were evident in toileting, stair climbing, dressing, and ambulation, among others. The FIM highlighted differences in bladder and bowel controls. Finally, laboratory results revealed contrasting hemoglobin levels, hematocrit levels, and platelet counts between the groups.

The distributions of patient data across the different datasets from the two hospitals are shown in Table 2. Notably, the “Lab” dataset was the most voluminous of all the datasets. Nevertheless, all the datasets were rigorously evaluated to optimize the machine learning models. The data revealed a marked difference in the composition of the datasets between the two hospitals. Primary analysis was performed using the comprehensive “Lab + ISNCSCI + K-MBI + FIM” dataset from DKUH. To increase the precision of external validation, distinct training and validation datasets were derived from the “Lab + ISNCSCI” collection of both DKUH and CNUH, thus facilitating cross-institutional validation. DKUH evaluated the performance of machine learning models using different dataset combinations, such as “Lab”, “Lab + ISNCSCI”, “Lab + ISNCSCI + K-MBI”, and “Lab + ISNCSCI + K-MBI + FIM”. In contrast, the CNUH evaluations focused primarily on the “Lab”, “ISNCSCI”, and “Lab + ISNCSCI” datasets due to significant data gaps in the K-MBI and FIM metrics.

### 3.3. Predictive Performance of the Machine Learning Models

Table 3 delineates the predictive properties of various machine learning (ML) models across distinct dataset combinations at Dankook University Hospital (DKUH). A comparison across identical algorithms revealed that the “Lab + ISNCS + CI + K-MBI + FIM” amalgam dataset consistently surpassed the other datasets in terms of AUC values. Within the same dataset, algorithmic variations demonstrated different levels of performance; the GNN-GCN algorithm outperformed the “Lab” dataset, and the KNN algorithm outperformed the others in the “Lab + ISNCSCI” dataset, whereas the SVM algorithm consistently stood out in the “Lab + ISNCSCI + K-MBI” and “Lab + ISNCSCI + K-MBI + FIM” datasets. Remarkably, the SVM_linear algorithm in the “Lab + ISNCSCI + K-MBI + FIM” dataset achieved a pinnacle AUC of 0.904, an accuracy of 0.944, and an F1-score of 0.907.

Feature selection was used strategically to improve the predictive accuracy of our models. Figure 2 shows the t-SNE plots before and after this important feature selection process. As shown, the demarcation between the non-PU and PU groups became much more pronounced after feature selection, illustrating the critical role of feature selection in achieving clearer data differentiation.

Figure 3 shows the importance scores of the top 39 parameters identified by the SVM_linear model. These parameters are spread across several categories, including Lab, ISNCSCI, K-MBI, and FIM. The representation of each category in this ranking indicates its integral role in influencing the model’s predictions. The most prominent parameter in this evaluation was “Ambulation” in the K-MBI, which received the highest importance score of 1.212, followed by lower body dressing, transfer to bed, chair, and wheelchair, eating, and bathing in the FIM. This score demonstrated the importance of the functional status of the upper and lower extremities in the modeling process. In addition, the balanced presence of clinical scales such as the K-MBI and FIM, combined with laboratory and neurological data from Lab and ISNCSCI, highlights the variety of factors considered important in this model. This combination of factors demonstrates the intricate blend of physical, cognitive, and biological considerations that the model accounts for when analyzing outcomes.

In the CNUH dataset, Table 4 describes the predictive effectiveness of various machine learning models across different combinations of datasets and algorithms. Within each algorithm, the “ISNCSCI” dataset predominantly registered the highest AUC for the GNN-GCN, DNN, KNN, and random forest algorithms. Conversely, the “Lab + ISNCSCI” dataset included the SVM_linear, SVM_RBF, and logistic regression algorithms. When comparing different algorithms on the same dataset, the KNN algorithm consistently had the highest AUC in all three datasets. Overall, the KNN algorithm showed the best predictive performance on the “ISNCSCI” dataset, achieving an AUC of 0.737, an accuracy of 0.891 and an F1-score of 0.661.

The t-SNE plot before and after feature selection in the CNUH model is shown in Appendix A. The non-PU and PU groups were more clearly classified after feature selection. The changes in the distribution patterns of the non-PU and PU groups after feature selection indicate notable changes. The eleven feature parameters were ranked by importance scores based on the outcome of the KNN model (Appendix A), and the ASIA impairment scale (AIS) item of the ISNCSCI had the highest importance score of 0.19.

Figure 4 shows the receiver operating characteristic (ROC) curves for the optimal datasets from both DKUH and CNUH. With respect to the DKUH dataset, which comprises the Lab + ISNCSCI + K-MBI + FIM variables, the SVM_linear algorithm distinctly surpassed the other methods, with an AUC of 0.904, an accuracy of 94.4%, a sensitivity of 0.840, a specificity of 0.968, and an F1-score of 0.907. The CNUH dataset, which was based exclusively on ISNCSCI variables, exhibited more uniform results across different algorithms. Notably, the KNN algorithm had an AUC of 0.737, an accuracy of 89.1%, a sensitivity of 0.562, a specificity of 0.913, and an F1-score of 0.661. Overall, the performance at CNUH was more restrained than that at DKUH. The SVM_linear algorithm maintained its superior performance with only the ISNCSCI variables in the DKUH dataset, but its AUC and accuracy were considerably lower than those of the combination of the Lab + ISNCSCI + K-MBI + FIM variables and even the CNUH results (Appendix A).

Figure 5 shows the decision tree model employed to distinguish between the Non-PU and PU groups. The primary discriminator is “K-MBI: ambulation”, with a threshold value of 2.338. Subjects who scored below this threshold were predominantly categorized using subsequent discriminators, notably “ISNCSCI: motor score of Rt. lower extremity” (≤18.314) and “FIM: eating” (≤5.033). In contrast, for those surpassing the “K-MBI: ambulation” threshold, “FIM: walk/wheelchair” (≤3.445) and “ISNCSCI: motor score of Rt. lower extremity” (≤7.388) emerged as salient discriminators. The tree further expands to encompass laboratory parameters such as platelet count, mean corpuscular hemoglobin, and eosinophil count, as well as multiple neurological and functional metrics. Each branch point denotes a unique criterion that aids in efficiently classifying subjects into the non-PU and PU groups.

## 4. Discussion

In our quest to improve outcomes for patients with SCI, our study’s application of ML techniques marks a shift from traditional areas of focus, including neurological and functional outcomes [30,31,32,33,34,35], to the proactive prevention of PUs. These prevalent yet preventable complications have a profound impact on the recovery and quality of life of patients with SCI [36]. Our innovative use of ML, ranging from SVM to DNN to GNN, has enabled us to delve into complex datasets and extract critical insights from nonlinear relationships for more accurate PU predictive modeling. This methodology underscores the potential of ML to go beyond the boundaries of conventional statistical analysis.

A key finding of our study was the differential performance of the linear SVM model across different datasets. The DKUH dataset, with its larger sample size and diverse baseline characteristics such as age range, injury severity, and neurological status, provided a different context for ML application than the CNUH dataset. This contrast in performance underscores the influence of specific dataset attributes on the success of ML models and highlights the need for data that encapsulate a broad range of patient scenarios to improve predictive accuracy in diverse clinical settings.

Furthermore, our analysis revealed the paramount importance of functional parameters, including walking (K-MBI), lower body dressing, transfers, eating and bathing (FIM), in predicting PUs (Figure 3). This finding highlights the dominance of functional data over neurological factors in risk assessment. The need for functional assessment is particularly pronounced in conditions such as spinal shock, where the neurological status may be uncertain. In addition to these functional indicators, our study also draws attention to the importance of pre-onset laboratory markers related to inflammation and anemia, such as lymphocyte, neutrophil and eosinophil counts, as well as MCHC and RBC counts, which is consistent with the findings of a previous study [37]. Although not primary predictors, their association with increased PU risk is consistent with previous research and underscores their importance in PU risk assessment.

As we move toward clinical application, we have developed a decision tree algorithm based on the results of our study. This algorithm incorporates key parameters identified as significant in predicting PUs, such as functional status indicators (e.g., mobility and self-care ability) and relevant laboratory markers (e.g., inflammatory, and hematological parameters). Designed as a user-friendly decision support tool, it systematically evaluates these factors to estimate PU risk, providing clinicians with a structured framework for early intervention. While the algorithm is promising, extensive validation in diverse clinical settings is essential to determine its utility and efficacy. Our preliminary external validation efforts have revealed variability in performance across datasets from different institutions, underscoring the challenges of creating a universally applicable ML-based prediction tool. These observations not only highlight the intricacies of ML model generalization, but also pave the way for further research to refine and adapt the algorithm for broader clinical use [38,39].

The limitations of our study are openly acknowledged, particularly with respect to the limited sample size and the brevity of the observation period. We recognize the potential of expansive data sources such as the National Spinal Cord Injury Statistical Center (NSCISC; https://www.nscisc.uab.edu/, accessed on 19 October 2021) and the National Institutes of Health (NIH) National Institute of Neurological Disorders and Stroke (NINDS; https://www.commondataelements.ninds.nih.gov/Spinal%20Cord%20Injury, accessed on 19 October 2021), although their use was limited by their mismatch with the acute and subacute phase specificity of our research, particularly the lack of time-sensitive laboratory data relevant to the onset of PUs. In addition, the design of our study needed a rigorous selection process to include only participants with unique records, limiting our dataset.

We envision that future studies include a wider network, integrate data from multiple centers, and account for the temporal progression of PUs. The exploration of hybrid machine learning frameworks that combine the strengths of different algorithms may hold the key to improving predictive accuracy. Our ultimate goal is to develop a reliable predictive framework that will not only facilitate the prevention and early treatment of PUs in clinical settings, but also have a tangible impact on the care and quality of life of patients with SCI. This framework will enable clinicians to intervene more effectively, potentially reducing the incidence of PUs and their associated complications. By improving early detection and intervention strategies, we aim to contribute to better health outcomes, increased independence, and overall well-being for patients with SCI.

## 5. Conclusions

In this study, we successfully developed a prediction model for PUs after SCI during the acute and subacute stages of the hospital stay using an ML algorithm, especially the SVM linear model. Our findings underscore the critical role of functional data, in addition to neurological and laboratory data, in the development of effective PU prediction models. Specifically, the five most important functional parameters identified were ambulation, lower body dressing, bed/chair/wheelchair transfers, eating, and bathing. These parameters, which are indicative of a patient’s mobility and self-care capabilities, are critical in predicting PU risk. The integration of these functional aspects into our machine learning-driven models holds great promise for the early detection and prevention of PUs in clinical settings, potentially leading to improved patient care and outcomes for patients with SCI.

## Figures and Tables

**Figure 1 jcm-13-00990-f001:**
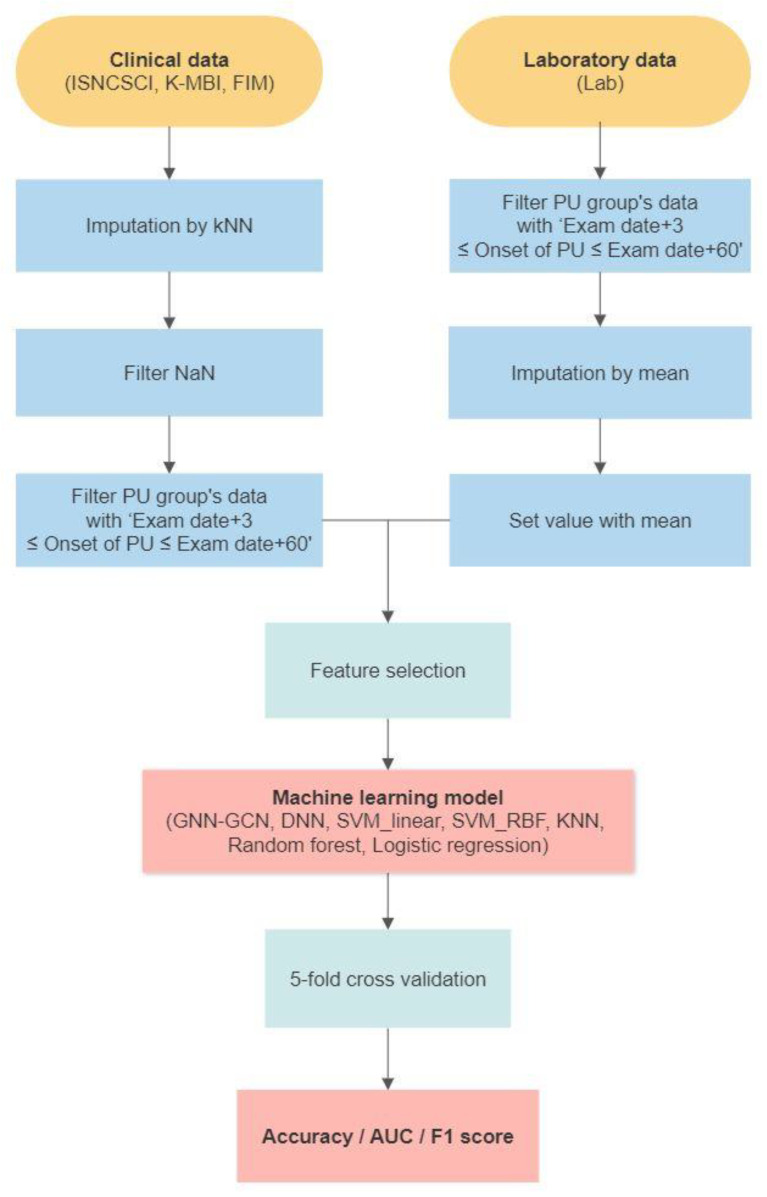
Flow of the machine learning process.

**Figure 2 jcm-13-00990-f002:**
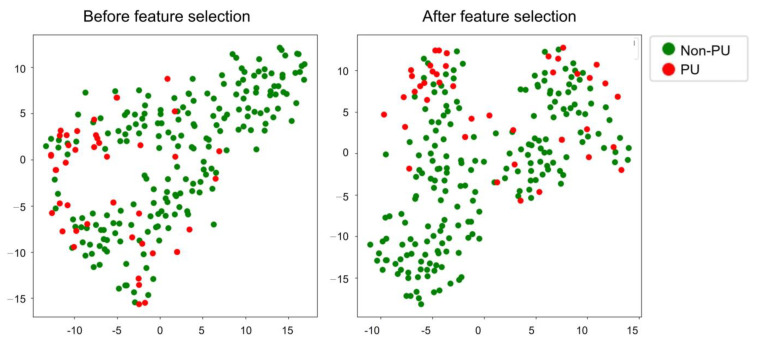
t-SNE plot before (**left**) and after (**right**) feature selection in the DKUH data.

**Figure 3 jcm-13-00990-f003:**
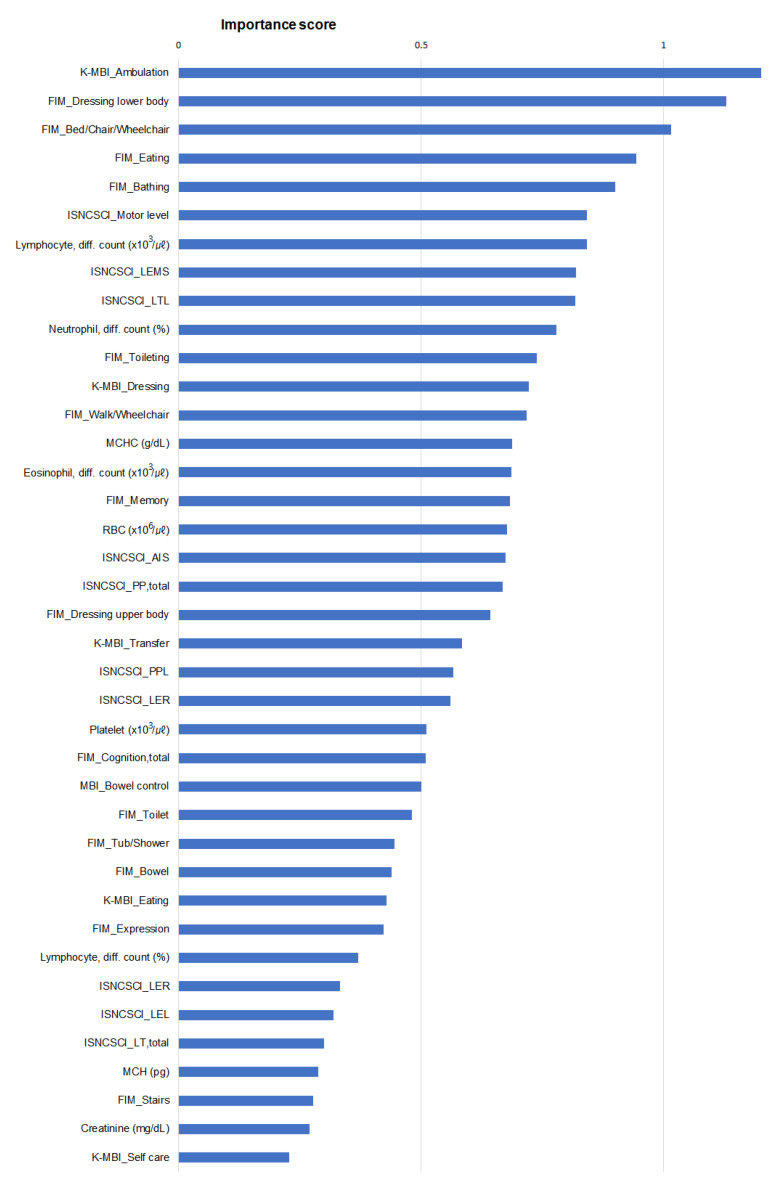
Importance scores of the top 39 featured parameters based on the outcome of the SVM_linear model. Abbreviations: K-MBI = Korean version of the Modified Barthel Index; FIM = Functional Independence Measure; ISNCSCI = International Standards for Neurological Classification of Spinal Cord Injury; LEMS = lower extremity, motor subscore; LTL = light touch, left; MCHC = mean corpuscular hemoglobin concentration; RBC = red blood cell; AIS = ASIA impairment scale; PPL = pinprick, left; LER = lower extremity, right; LEL = lower extremity, left; LT, total = light touch, total; MCH = mean corpuscular hemoglobin.

**Figure 4 jcm-13-00990-f004:**
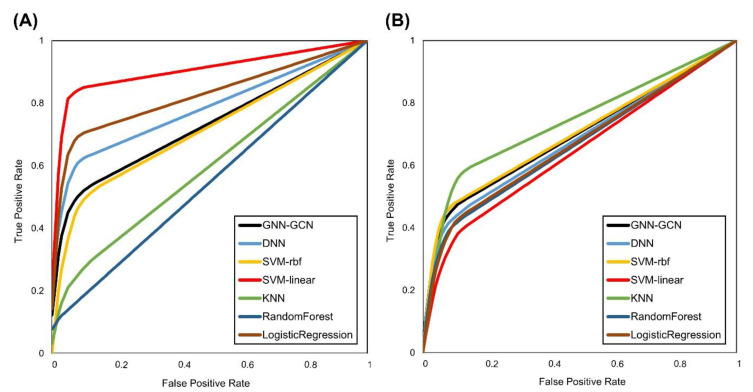
Receiver operating characteristic (ROC) curve of each machine learning algorithm in (**A**) DKUH using the Lab + ISNCSCI + K-MBI + FIM dataset and (**B**) CNUH using the ISNCSCI dataset.

**Figure 5 jcm-13-00990-f005:**
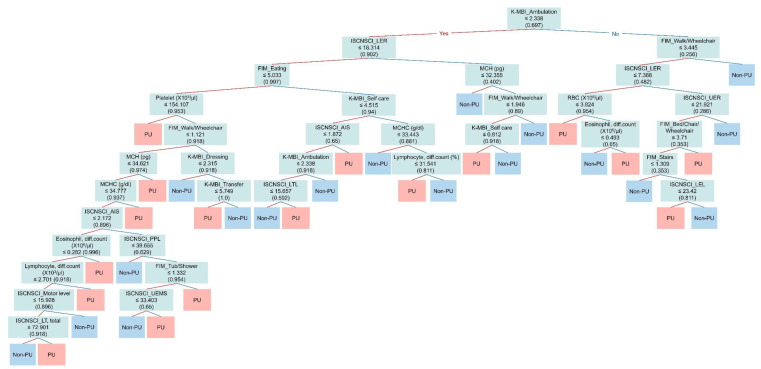
Decision tree of the SVM_linear model trained with the “Lab + ISNCSCI + K-MBI + FIM” dataset of DKUH to classify the non-PU and PU groups. The “ambulation” subscale of the K-MBI was identified as the first single discriminator for determination of the two groups. The red line indicates “yes”, and the blue line indicates “no”. Abbreviations: K-MBI = Korean version of the Modified Barthel Index; ISNCSCI = International Standards for Neurological Classification of Spinal Cord Injury; FIM = Functional Independence Measure; LER = lower extremity, right; UER = upper extremity, right; AIS = ASIA Impairment Scale; LTL = light touch, left; LEL = lower extremity, left; PPL = pinprick, left; UEMS = Upper Extremity Motor Subscore; LT, total = light touch, total, MCH = mean corpuscular hemoglobin; MCHC = mean corpuscular hemoglobin concentration.

**Table 1 jcm-13-00990-t001:** Baseline characteristics of patients with and without pressure ulcers at DKUH and CNUH.

Parameters	DKUH (*n* = 238)			CNUH (*n* = 385)		
	Non-PU (*n* = 199)	PU (*n* = 39)	*p* Value	Non-PU (*n* = 362)	PU (*n* = 23)	*p* Value
Baseline characteristics						
Sex (male)	158 (79.4%)	31 (79.5%)	0.99	247 (68.2%)	19 (82.6%)	0.148
Age	54.22 ± 14.26	48.95 ± 17.02	0.042 *	58.51 ± 15.62	57.00 ± 17.91	0.657
Height	166.85 ± 7.71	167.85 ± 7.51	0.867	165.00 ± 9.06	166.55 ± 9.54	0.495
Weight	65.18 ± 10.59	65.17 ± 11.66	0.996	64.55 ± 12.38	63.41 ± 9.89	0.680
Alcohol consumption	99 (50.0%)	26 (66.7%)	0.057	103 (28.6%)	6 (26.1%)	0.795
Smoking status	73 (36.9%)	16 (41.0%)	0.624	97 (27.0%)	5 (21.7%)	0.579
Diabetes mellitus	37 (18.7%)	8 (20.5%)	0.79	67 (18.5%)	4 (17.4%)	0.893
Hypertension	60 (30.3%)	13 (33.3%)	0.708	130 (35.9%)	8 (34.8%)	0.913
Neurologic disease	34 (17.1%)	4 (10.5%)	0.313	16 (4.4%)	0 (0.0%)	0.303
Cardiovascular disease	2 (1.0%)	3 (7.7%)	0.008 *	38 (10.5%)	0 (0.0%)	0.102
Pulmonary disease	9 (4.5%)	4 (10.3%)	0.15	25 (6.9%)	2 (8.7%)	0.745
Clinical parameters						
Hospital days	53.49 ± 31.87	96.72 ± 67.15	0.000 *	68.15 ± 42.92	118.17 ± 81.66	0.008 *
Braden scale	15.48 ± 3.62	13.69 ± 2.45	0.000 *	Null	Null	Null
Traumatic injury	158 (79.8%)	33 (84.6%)	0.487	199 (55.0%)	16 (69.6%)	0.172
Mechanism of injury						
Traffic accident	67 (37.6%)	14 (40.0%)	0.581	59 (30.1%)	5 (31.3%)	0.183
Falls	58 (32.6%)	12 (34.3%)		48 (24.5%)	8 (50.0%)	
Hit by falling objects	12 (6.7%)	4 (11.4%)		7 (3.6%)	0 (0.0%)	
Sports	0 (0%)	0 (0%)		2 (1.0%)	0 (0.0%)	
Others	41 (23.0%)	5 (14.3%)		80 (40.8%)	3 (18.8%)	
Combined injury	42 (21.1%)	13 (33.3%)	0.098	56 (15.5%)	6 (26.1%)	0.179
Number of operations	1.13 ± 0.741	1.51 ± 1.048	0.035 *	0.82 ± 1.01	0.61 ± 0.84	0.333
Total time of operations (min)	229.58 ± 146.85	256.77 ± 164.36	0.301	269.18 ± 176.42	245.67 ± 85.33	0.692
GCS total	14.69 ± 0.92	14.06 ± 2.76	0.212	14.00 ± 2.68	12.00 ± 1.41	0.337
GCS Eye	3.83 ± 0.44	3.63 ± 0.942	0.236	3.75 ± 0.87	4.00 ± 0.00	0.700
GCS Motor	5.93 ± 0.30	5.66 ± 0.90	0.098	5.83 ± 0.58	6.00 ± 0.00	0.700
GCS Verbal	4.92 ± 0.64	4.78 ± 1.52	0.605	4.50 ± 1.17	2.00 ± 1.41	0.018 *
ISNCSCI						
ASIA impairment scale (AIS)						
A	12 (6.3%)	14 (36.8%)	0.000 *	23 (7.6%)	7 (31.8%)	0.000 *
B	9 (4.8%)	5 (13.2%)		8 (2.6%)	4 (18.2%)	
C	27 (14.3%)	12 (31.6%)		50 (16.5%)	8 (36.4%)	
D	138 (73.0%)	7 (18.4%)		221 (72.9%)	3 (13.6%)	
E	3 (1.6%)	0 (0%)		1 (0.3%)	0 (0.0%)	
NLI (neurologic level of injury)	9.60 ± 8.14	10.31 ± 7.14	0.612	10.52 ± 8.57	8.87 ± 6.23	0.241
Motor						
Motor level	10.54 ± 8.66	10.46 ± 7.07	0.95	11.90 ± 8.72	8.96 ± 6.17	0.040 *
UER	20.06 ± 6.68	18.15 ± 8.51	0.193	20.07 ± 5.21	14.91 ± 9.00	0.012 *
UEL	19.13 ± 6.97	18.59 ± 8.15	0.667	20.07 ± 5.21	15.30 ± 8.77	0.017 *
UEMS	39.07 ± 12.75	36.74 ± 16.54	0.41	40.15 ± 9.93	30.22 ± 17.65	0.014 *
LER	18.22 ± 8.71	6.21 ± 8.08	0.000 *	16.18 ± 7.42	5.96 ± 7.97	0.000 *
LEL	17.84 ± 8.63	6.32 ± 8.59	0.000 *	16.13 ± 7.49	5.39 ± 7.67	0.000 *
LEMS	35.72 ± 16.71	12.50 ± 16.48	0.000 *	32.31 ± 14.40	11.35 ± 15.54	0.000 *
Motor score, total	74.76 ± 22.42	48.89 ± 20.97	0.000 *	72.45 ± 18.69	41.57 ± 28.48	0.000 *
Sensory						
Sensory level	14.19 ± 10.44	12.69 ± 7.36	0.285	12.07 ± 9.30	11.83 ± 7.14	0.875
LTR	45.76 ± 10.40	39.62 ± 12.57	0.001 *	39.17 ± 11.06	34.04 ± 11.40	0.032 *
LTL	45.62 ± 10.61	39.15 ± 12.49	0.001 *	39.28 ± 11.11	33.70 ± 10.87	0.020 *
LT, total	91.38 ± 20.56	78.77 ± 25.00	0.001 *	78.44 ± 21.92	67.74 ± 22.15	0.024 *
PPR	45.72 ± 10.28	40.08 ± 12.29	0.003 *	38.38 ± 11.54	33.83 ± 12.64	0.069
PPL	45.90 ± 10.59	39.82 ± 12.17	0.002 *	38.80 ± 11.42	33.61 ± 11.94	0.036 *
PP, total	91.62 ± 20.35	79.90 ± 24.42	0.002 *	77.17 ± 22.51	67.43 ± 24.44	0.046 *
Sensory score, total	183.00 ± 40.56	158.67 ± 49.18	0.001 *	155.61 ± 43.84	135.17 ± 45.96	0.031 *
K-MBI						
Self-care	2.90 ± 1.90	2.33 ± 1.95	0.088 *	3.12 ± 1.72	2.33 ± 1.97	0.308
Bathing	1.79 ± 1.61	0.64 ± 0.84	0.000 *	2.15 ± 1.64	1.67 ± 1.51	0.502
Feeding	5.87 ± 3.91	5.10 ± 4.27	0.268	6.12 ± 3.60	6.00 ± 4.73	0.941
Toileting	4.05 ± 3.63	1.31 ± 1.45	0.000 *	4.44 ± 3.41	1.83 ± 1.84	0.016 *
Stair climbing	1.62 ± 3.04	0.05 ± 0.32	0.000 *	1.56 ± 2.87	0.00 ± 0.00	0.001 *
Dressing	4.23 ± 3.29	2.36 ± 2.08	0.000 *	5.12 ± 3.05	1.83 ± 1.84	0.014 *
Bowel management	6.33 ± 4.13	2.51 ± 3.53	0.000 *	6.83 ± 4.07	2.33 ± 3.88	0.015 *
Bladder management	5.04 ± 4.70	1.08 ± 2.93	0.000 *	5.76 ± 4.12	1.17 ± 2.04	0.001 *
Ambulation	4.91 ± 5.40	0.69 ± 1.15	0.000 *	5.07 ± 4.60	1.50 ± 1.64	0.002 *
Transfer	6.79 ± 5.42	2.08 ± 2.26	0.000 *	7.27 ± 5.08	3.67 ± 3.62	0.102
Total	43.53 ± 30.17	18.15 ± 14.92	0.000 *	46.19 ± 27.48	22.33 ± 17.93	0.044 *
FIM						
Eating	4.13 ± 2.28	3.87 ± 2.54	0.556	4.50 ± 2.12	4.00 ± 4.24	0.773
Grooming	3.94 ± 2.21	3.33 ± 2.13	0.114	3.56 ± 1.92	2.50 ± 2.12	0.472
Bathing	2.63 ± 1.63	1.64 ± 0.78	0.000 *	2.94 ± 1.59	1.50 ± 0.71	0.228
Dressing upper body	3.49 ± 1.94	2.67 ± 1.71	0.015 *	3.72 ± 1.97	4.00 ± 4.24	0.865
Dressing lower body	3.07 ± 1.91	1.59 ± 0.94	0.000 *	3.39 ± 1.88	2.00 ± 1.41	0.330
Toileting	3.03 ± 2.00	1.64 ± 0.84	0.000 *	3.50 ± 1.95	1.50 ± 0.71	0.175
Self-care, total	20.29 ± 10.84	14.74 ± 7.72	0.000 *	21.61 ± 10.86	15.50 ± 13.44	0.466
Bladder control	3.88 ± 2.76	1.59 ± 1.70	0.000 *	4.56 ± 2.41	1.00 ± 0.00	0.000 *
Bowel control	4.46 ± 2.50	2.31 ± 1.94	0.000 *	4.78 ± 2.37	1.50 ± 0.71	0.009 *
Sphincter control, total	8.34 ± 4.88	3.90 ± 3.39	0.000 *	9.33 ± 4.72	2.50 ± 0.71	0.000 *
Transfer to bed/chair/wheelchair	3.33 ± 2.04	1.74 ± 0.85	0.000 *	3.06 ± 1.73	2.00 ± 1.41	0.420
Transfer to toilet	3.09 ± 2.04	1.51 ± 0.68	0.000 *	3.00 ± 1.78	1.50 ± 0.71	0.263
Transfer to tub/shower	2.93 ± 1.96	1.49 ± 0.64	0.000 *	2.83 ± 1.76	1.50 ± 0.71	0.311
Locomotion with walk/wheelchair	3.02 ± 1.93	1.44 ± 0.64	0.000 *	2.89 ± 1.64	2.00 ± 1.41	0.474
Locomotion to stairs	1.88 ± 1.67	1.05 ± 0.22	0.000 *	2.00 ± 1.75	1.00 ± 0.00	0.440
Transfer/Locomotion, total	14.25 ± 9.15	7.23 ± 2.72	0.000 *	13.78 ± 8.16	8.00 ± 4.24	0.345
Comprehension	6.86 ± 0.61	6.72 ± 0.916	0.344	6.17 ± 1.76	4.50 ± 3.54	0.255
Expression	6.84 ± 0.66	6.69 ± 0.83	0.304	6.22 ± 1.73	4.50 ± 3.54	0.235
Social interaction	6.81 ± 0.78	6.67 ± 1.01	0.324	6.28 ± 1.64	4.50 ± 3.54	0.201
Problem solving	6.78 ± 0.81	6.67 ± 1.11	0.46	6.17 ± 1.76	4.50 ± 3.54	0.255
Memory	6.79 ± 0.74	6.72 ± 0.97	0.58	6.22 ± 1.73	4.50 ± 3.54	0.235
Cognition, total	33.98 ± 3.97	32.74 ± 6.36	0.248	31.06 ± 8.59	22.50 ± 17.68	0.235
FIM, total	76.86 ± 23.31	58.62 ± 13.85	0.000 *	75.78 ± 26.96	48.50 ± 36.06	0.201
Laboratory parameters						
White blood cells (×10^3^/µL)	8.87 ± 2.07	9.25 ± 2.13	0.301	6.95 ± 1.71	7.40 ± 2.53	0.234
Red blood cells (×10^6^/µL)	4.08 ± 0.39	3.83 ± 0.46	0.000 *	4.10 ± 0.47	3.96 ± 0.51	0.150
Hemoglobin (g/dL)	12.62 ± 1.26	12.06 ± 1.49	0.014 *	12.57 ± 1.38	11.82 ± 1.33	0.012 *
Hematocrit (%)	37.42 ± 3.49	35.35 ± 4.20	0.001 *	37.40 ± 3.85	35.44 ± 3.90	0.018 *
Mean corpuscular volume (fl)	91.77 ± 4.42	92.32 ± 4.22	0.471	91.41 ± 3.88	89.89 ± 4.81	0.074
Mean corpuscular hemoglobin (pg)	30.95 ± 1.66	31.50 ± 1.70	0.060	30.68 ± 1.52	29.91 ± 1.61	0.019 *
Mean corpuscular hemoglobin concentration (g/dL)	33.72 ± 0.75	34.13 ± 0.88	0.003 *	33.57 ± 0.69	33.29 ± 0.76	0.060
Platelets (×10^3^/µL)	240.29 ± 56.55	211.94 ± 67.70	0.006 *	256.03 ± 63.97	284.55 ± 82.67	0.043 *
Neutrophils, diff. count (%)	60.50 ± 8.85	64.46 ± 8.60	0.016 *	61.70 ± 7.13	64.88 ± 9.49	0.060
Lymphocytes, diff. count (%)	28.24 ± 7.71	24.38 ± 7.31	0.004 *	27.26 ± 6.53	23.59 ± 8.75	0.011
Monocytes, diff. count (%)	7.48 ± 1.88	7.51 ± 1.80	0.938	7.37 ± 1.52	7.65 ± 1.69	0.394
Eosinophils, diff. count (%)	3.33 ± 1.84	3.40 ± 2.29	0.842	3.07 ± 1.61	2.87 ± 1.51	0.578
Basophils, diff. count (%)	0.45 ± 0.26	0.45 ± 0.31	0.985	0.51 ± 0.19	0.53 ± 0.44	0.792
Neutrophils, diff. count (×10^3^/µL)	4.46 ± 1.58	5.23 ± 2.14	0.010 *	4.47 ± 1.42	5.09 ± 2.41	0.238
Lymphocytes, diff. count (×10^3^/µL)	1.84 ± 0.53	1.76 ± 0.61	0.420	1.74 ± 0.49	5.09 ± 2.41	0.028 *
Monocytes, diff. count (×10^3^/µL)	0.52 ± 0.17	0.56 ± 0.16	0.149	0.50 ± 0.14	0.55 ± 0.19	0.222
Eosinophils, diff. count (×10^3^/µL)	0.21 ± 0.11	0.24 ± 0.18	0.169	0.19 ± 0.11	0.19 ± 0.10	0.694
Basophils, diff. count (×10^3^/µL)	0.03 ± 0.02	0.03 ± 0.02	0.470	0.03 ± 0.01	0.04 ± 0.03	0.602
Creatinine (mg/dL)	0.71 ± 0.20	0.71 ± 0.27	0.902	2.25 ± 5.50	0.85 ± 1.03	0.000 *
Blood urea nitrogen (mg/dL)	16.11 ± 3.87	17.44 ± 5.61	0.072	15.23 ± 10.68	13.57 ± 4.23	0.461

Note: Values are presented as the number of subjects (%) or means ± standard deviations. The *p* values of the non-PU and PU groups were determined by the chi-squared test and independent *t* test; * *p* < 0.05. Abbreviations: PU = pressure ulcer, ISNCSCI = International Standards for Neurological Classification of Spinal Cord Injury; K-MBI = Korean version of the Modified Barthel Index; FIM = Functional Independence Measure.

**Table 2 jcm-13-00990-t002:** Number of patients in the dataset category.

Dataset	DKUH			CNUH		
	Non-PU	PU	Total	Non-PU	PU	Total
Lab	328	159 (253)	487	434	73 (92)	507
ISNCSCI	221	46 (55)	267	362	23 (24)	385
K-MBI	259	46 (59)	307	62	6 (6)	68
FIM	250	46 (59)	298	31	2 (2)	33
ISNCSCI + K-MBI	208	46 (48)	248	41	3 (3)	44
ISNCSCI + K-MBI + FIM	200	46 (46)	239	16	1 (1)	17
Lab + ISNCSCI	216	46 (55)	262	362	23 (24)	385
Lab + ISNCSCI + K-MBI	207	46 (48)	247	41	3 (3)	44
Lab + ISNCSCI + K-MBI + FIM	199	39 (46)	238	16	1 (1)	17

The numbers of patients satisfying each dataset in the two hospitals are presented. The numbers in parentheses represent the counts of PU events. Abbreviations: DKUH = Dankook University Hospital; CNUH = Chungnam National University Hospital; PU = pressure ulcer; Lab = laboratory data; ISNCSCI = International Standards for Neurological Classification of Spinal Cord Injury; K-MBI = Korean version of the Modified Barthel Index; FIM = Functional Independence Measure.

**Table 3 jcm-13-00990-t003:** Performance comparison of machine learning algorithms in each dataset from DKUH.

Model	Measure	Dataset			
		Lab	Lab + ISNCSCI	Lab + ISNCSCI + K-MBI	Lab + ISNCSCI + K-MBI + FIM
GNN-GCN	Sensitivity	0.442 ± 0.143	0.367 ± 0.190	0.508 ± 0.107	0.494 ± 0.163
	Specificity	0.883 ± 0.034	0.886 ± 0.068	0.913 ± 0.043	0.960 ± 0.036
	Accuracy	0.808 ± 0.052	0.788 ± 0.041	0.837 ± 0.040	0.873 ± 0.040
	AUC	0.656 ± 0.077	0.626 ± 0.078	0.710 ± 0.058	0.727 ± 0.082
	F1-score	0.662 ± 0.084	0.622 ± 0.076	0.720 ± 0.064	0.754 ± 0.085
DNN	Sensitivity	0.420 ± 0.192	0.132 ± 0.101	0.472 ± 0.135	0.600 ± 0.129
	Specificity	0.903 ± 0.023	0.966 ± 0.031	0.920 ± 0.031	0.963 ± 0.035
	Accuracy	0.834 ± 0.040	0.810 ± 0.030	0.836 ± 0.034	0.895 ± 0.040
	AUC	0.647 ± 0.090	0.549 ± 0.053	0.696 ± 0.069	0.781 ± 0.069
	F1-score	0.662 ± 0.105	0.545 ± 0.076	0.707 ± 0.067	0.808 ± 0.071
SVM_linear	Sensitivity	0.106 ± 0.169	0.245 ± 0.160	0.560 ± 0.132	0.840 ± 0.110
	Specificity	0.898 ± 0.008	0.945 ± 0.035	0.893 ± 0.034	0.968 ± 0.026
	Accuracy	0.818 ± 0.015	0.813 ± 0.035	0.830 ± 0.039	0.944 ± 0.031
	AUC	0.532 ± 0.056	0.595 ± 0.077	0.727 ± 0.071	0.904 ± 0.058
	F1-score	0.502 ± 0.087	0.599 ± 0.104	0.723 ± 0.067	0.907 ± 0.052
SVM_RBF	Sensitivity	0.298 ± 0.144	0.278 ± 0.130	0.525 ± 0.139	0.492 ± 0.165
	Specificity	0.882 ± 0.024	0.935 ± 0.040	0.893 ± 0.040	0.930 ± 0.039
	Accuracy	0.798 ± 0.037	0.811 ± 0.037	0.824 ± 0.044	0.847 ± 0.043
	AUC	0.585 ± 0.062	0.606 ± 0.066	0.709 ± 0.075	0.711 ± 0.084
	F1-score	0.590 ± 0.077	0.619 ± 0.079	0.709 ± 0.072	0.723 ± 0.085
KNN	Sensitivity	0.208 ± 0.195	0.432 ± 0.148	0.282 ± 0.128	0.246 ± 0.127
	Specificity	0.894 ± 0.018	0.898 ± 0.059	0.893 ± 0.047	0.811 ± 0.043
	Accuracy	0.813 ± 0.031	0.810 ± 0.049	0.779 ± 0.044	0.811 ± 0.043
	AUC	0.559 ± 0.074	0.665 ± 0.074	0.588 ± 0.067	0.594 ± 0.067
	F1-score	0.551 ± 0.103	0.670 ± 0.074	0.593 ± 0.074	0.605 ± 0.086
Random Forest	Sensitivity	0.122 ± 0.131	0.262 ± 0.144	0.208 ± 0.133	0.073 ± 0.074
	Specificity	0.899 ± 0.008	0.933 ± 0.043	0.953 ± 0.042	0.990 ± 0.019
	Accuracy	0.820 ± 0.015	0.807 ± 0.030	0.813 ± 0.030	0.818 ± 0.021
	AUC	0.532 ± 0.040	0.597 ± 0.062	0.581 ± 0.058	0.532 ± 0.038
	F1-score	0.510 ± 0.068	0.602 ± 0.073	0.584 ± 0.074	0.511 ± 0.065
Logistic Regression	Sensitivity	0.380 ± 0.196	0.352 ± 0.153	0.438 ± 0.144	0.683 ± 0.163
	Specificity	0.907 ± 0.017	0.942 ± 0.040	0.918 ± 0.036	0.964 ± 0.029
	Accuracy	0.838 ± 0.031	0.831 ± 0.041	0.828 ± 0.036	0.911 ± 0.033
	AUC	0.630 ± 0.084	0.647 ± 0.077	0.678 ± 0.072	0.823 ± 0.078
	F1-score	0.643 ± 0.104	0.665 ± 0.083	0.689 ± 0.069	0.841 ± 0.065

Abbreviations: Lab = laboratory data; ISNCSCI = International Standards for Neurological Classification of Spinal Cord Injury; K-MBI = Korean version of the Modified Barthel Index; FIM = Functional Independence Measure.

**Table 4 jcm-13-00990-t004:** Performance comparison of machine learning algorithms in each dataset of CNUH data.

Model	Measure	Dataset		
		Lab	ISNCSCI	Lab + ISNCSCI
GNN-GCN	Sensitivity	0.180 ± 0.195	0.426 ± 0.218	0.362 ± 0.200
	Specificity	0.923 ± 0.036	0.947 ± 0.035	0.945 ± 0.037
	Accuracy	0.877 ± 0.034	0.914 ± 0.034	0.908 ± 0.033
	AUC	0.551 ± 0.096	0.686 ± 0.107	0.653 ± 0.097
	F1-score	0.538 ± 0.078	0.666 ± 0.092	0.635 ± 0.087
DNN	Sensitivity	0.108 ± 0.134	0.398 ± 0.214	0.388 ± 0.194
	Specificity	0.944 ± 0.031	0.947 ± 0.036	0.949 ± 0.033
	Accuracy	0.892 ± 0.031	0.913 ± 0.035	0.914 ± 0.028
	AUC	0.526 ± 0.068	0.672 ± 0.105	0.669 ± 0.092
	F1-score	0.523 ± 0.066	0.654 ± 0.099	0.654 ± 0.077
SVM_linear	Sensitivity	0.124 ± 0.132	0.416 ± 0.204	0.418 ± 0.181
	Specificity	0.961 ± 0.035	0.950 ± 0.027	0.925 ± 0.038
	Accuracy	0.909 ± 0.035	0.889 ± 0.032	0.894 ± 0.036
	AUC	0.543 ± 0.069	0.643 ± 0.072	0.672 ± 0.089
	F1-score	0.547 ± 0.084	0.614 ± 0.065	0.636 ± 0.081
SVM_RBF	Sensitivity	0.140 ± 0.173	0.362 ± 0.153	0.420 ± 0.214
	Specificity	0.944 ± 0.028	0.924 ± 0.035	0.957 ± 0.028
	Accuracy	0.894 ± 0.028	0.917 ± 0.025	0.924 ± 0.025
	AUC	0.542 ± 0.086	0.683 ± 0.098	0.689 ± 0.103
	F1-score	0.537 ± 0.083	0.661 ± 0.086	0.676 ± 0.085
KNN	Sensitivity	0.340 ± 0.175	0.562 ± 0.236	0.538 ± 0.223
	Specificity	0.875 ± 0.040	0.913 ± 0.030	0.910 ± 0.029
	Accuracy	0.842 ± 0.037	0.891 ± 0.027	0.887 ± 0.028
	AUC	0.607 ± 0.085	0.737 ± 0.114	0.724 ± 0.109
	F1-score	0.559 ± 0.053	0.661 ± 0.068	0.651 ± 0.069
Random Forest	Sensitivity	0.066 ± 0.114	0.386 ± 0.191	0.378 ± 0.194
	Specificity	0.996 ± 0.008	0.943 ± 0.031	0.945 ± 0.033
	Accuracy	0.938 ± 0.010	0.909 ± 0.028	0.910 ± 0.033
	AUC	0.531 ± 0.056	0.665 ± 0.093	0.662 ± 0.101
	F1-score	0.533 ± 0.083	0.646 ± 0.080	0.649 ± 0.093
Logistic Regression	Sensitivity	0.148 ± 0.160	0.408 ± 0.184	0.442 ± 0.191
	Specificity	0.941 ± 0.035	0.926 ± 0.033	0.928 ± 0.033
	Accuracy	0.892 ± 0.032	0.895 ± 0.032	0.898 ± 0.032
	AUC	0.545 ± 0.078	0.667 ± 0.093	0.685 ± 0.094
	F1-score	0.538 ± 0.070	0.636 ± 0.083	0.648 ± 0.082

The KNN model trained with the “ISNCSCI” dataset from CNUH demonstrated the highest performance, with an AUC of 0.737. Abbreviations: Lab = laboratory data; ISNCSCI = International Standards for Neurological Classification of Spinal Cord Injury.

## Data Availability

The data presented in this study are available from the corresponding authors upon reasonable request.

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
