# Peer review of "Integrated Machine Learning Approach for the Early Prediction of Pressure Ulcers in Spinal Cord Injury Patients"

_jcm, 2024, doi:10.3390/jcm13040990_

Round 1

Reviewer 1 Report

Comments and Suggestions for Authors

It is an interesting topic which is rarely addressed.

However, there are many aspects that need to be improved:

Lines 51-53: ,, …the Braden Scale, Norton Scale, 51 and Spinal Cord Injury Pressure Ulcer Scale, although prevalent, offer limited predictive 52 accuracy with noticeable variability among individual SCI patients.

 You can specify that these scales are predominant in clinical use.

Lines 79-80: The primary goal of this study was to establish optimal prediction models based on 79 a comprehensive integration of clinical, physical and biological parameters.

You mentioned the main purpose of the study. Do you think you could consider secondary objectives? It would provide a more comprehensive picture of the objectives of the study.

Lines 92-94: ,,Patients were included if they underwent surgical or conservative treatment for traumatic or nontraumatic SCI with confirmed spinal cord signal changes by spinal magnetic resonance imaging (MRI) “

Can you briefly explain why confirmation of spinal cord signal changes is relevant or necessary to be included in the study?

Lines 95-97 : The clinical data of the patients during the initial hospitalization period for SCI were collected by three researchers who were not involved in the statistical analysis or development of the ML model in this study.

Can you provide a brief justification for why the researchers collecting the clinical data were not involved in the statistical analysis or model development? This helps establish separation of roles and adds transparency to the study process.

Lines 95 si 102: “The clinical data of the patients during the initial hospitalization period for SCI” and “during the initial hospitalization period”.

Ensuring consistency in the use of terminology. For example, you refer to "initial period of hospitalization for SCI" and "initial period of hospitalization." Make sure these terms are used interchangeably or clarify if there is a distinction.

Line 95: “from 1996 to 2022” 

Can you specify the period more clearly? (eg month?)

Lines 119 and 124-125: ensure consistency in the use of terminology - in line 119 you use "support vector machine (SVM) linear" and in lines 124-125 linear support vector machine (SVM_linear)

Lines 376-377: "The limitations of our study are openly acknowledged, particularly with respect to the limited sample size and the brevity of the observation period".

I appreciate you pointing out the limitations of the study regarding the sample size and the short duration of the observation period.

 My comments are only intended to make the paper better. Good luck! 

Reviewer 2 Report

Comments and Suggestions for Authors

This study aimed to develop predictive models for pressure ulcers (PUs) in spinal cord injury (SCI) patients using machine learning (ML) techniques. Analyzing the medical records of 539 SCI patients, they found a 35% incidence of PUs during hospitalization. The study considered 139 variables, including baseline characteristics, neurological status, and functional ability. Various ML methods were employed, and the linear support vector machine (SVM_linear) performed best, showing superior predictive ability (AUC = 0.904, accuracy = 0.944) through cross-validation. The critical factors for PU development were limb functional status and inflammation markers. The study underscores the importance of a multidimensional data approach for effective PU prediction in SCI patients, particularly in acute and subacute phases.

The study is of interest; however, there are some potential points for improvement:

Introduction

1.      Please provide more information about spinal cord injury (e.g., severity, characteristics of neurological and motor impairments) related to an incidence of pressure ulcers.

2.      Please provide more information about pressure ulcers (PUs), e.g., types, causes, common sites, problems, medical care, etc., that is linked to the importance of prediction.

3.      Please provide details of the systems, e.g., the Braden Scale, Norton Scale, and Spinal Cord Injury Pressure Ulcer Scale, in terms of how they are used and how they link to pressure ulcers.

4.      The text references various sources, but it would be beneficial to include specific citations within the text to enhance transparency and credibility. For example, citing the studies that support the prevalence of PUs among SCI patients or the limitations of existing assessment scales would reinforce the claims made.

5.      While the text mentions the severe consequences of untreated PUs, it could provide a more detailed explanation of the potential impact on patients' overall well-being, healthcare costs, and the healthcare system. This would add depth to our understanding of the significance of preventing and managing PUs.

6.      The text briefly mentions the limitations of existing assessment tools, such as the Braden Scale, Norton Scale, and Spinal Cord Injury Pressure Ulcer Scale. Expanding on these limitations and providing a comparative analysis would offer a more comprehensive view of the challenges faced by clinicians in predicting PUs.

7.      While molecular markers, including cytokine levels, are introduced as potential predictors of ulcer development, the text does not elaborate on how these markers work or their current status in clinical applications. Including a brief discussion on the promise and challenges of using molecular markers would add depth to this aspect.

8.      The text mentions the importance of cutting-edge technologies but does not specify the types of technologies or how they contribute to the prevention and management of PUs. Adding specific examples or brief explanations of relevant technologies would enhance clarity.

9.      The introduction mentions the emergence of ML in healthcare but lacks a clear transition to the specific focus on ML applications in spinal cord injury (SCI) and pressure ulcers (PUs). A smoother transition could enhance the flow of the text.

10.   The text mentions the growth of nuanced algorithms, including decision trees and deep learning models, but a brief explanation of how these algorithms work and their specific strengths in healthcare applications would improve understanding, especially for readers less familiar with ML.

Materials and methods

1.      While the text mentions various ML techniques employed in the study, it could benefit from a brief explanation of these techniques for readers unfamiliar with them. This would enhance understanding and engagement.

Results

1.      Please enlarge the font size in the Figure 5.

Discussion

1.      The text occasionally repeats certain points, such as the emphasis on functional data and the challenges of model generalizability. It is important to avoid redundancy and maintain a concise presentation of information.

2.      The text mentions differences in ML performance between datasets from different institutions, but it could provide more specifics on the characteristics of these datasets (e.g., size, demographics) to better understand the challenges faced during external validation.

3.      While the text mentions the development of a decision tree algorithm, it would be helpful to discuss the features or criteria that the decision tree considers and how it might be practically applied in clinical settings.

4.      The conclusion discusses the ultimate goal of establishing a reliable predictive framework, but it could also emphasize the potential practical implications of the study’s findings on patient care and quality of life for SCI patients.

Conclusion

1.      The text describes the five most important parameters derived from functional data, but incorporating their details.

Round 2

Reviewer 2 Report

Comments and Suggestions for Authors

Thank you for addressing the concerns raised in the previous review. This revision enhances the overall quality and readability of the manuscript.